# A pilot phase Ib/II study of whole-lung low dose radiation therapy (LDRT) for the treatment of severe COVID-19 pneumonia: First experience from Africa

**Mansoor Saleh** [1,2]*, **Karishma Sharma** [1], **Jasmit Shah** [1,3], **Farrok Karsan** [2], **Angela Waweru** [2], **Martin Musumbi** [3], **Reena Shah** [3], **Shahin Sayed** [4], **Innocent Abayo** [1], **Noureen Karimi** [1], **Stacey Gondi** [1], **Sehrish Rupani** [1], **Grace Kirathe** [1], **Heldah Amariati** [1]

**1** Clinical Research Unit, Aga Khan University Hospital, Nairobi, Kenya, **2** Department of Haematology and Oncology, Aga Khan University Hospital, Nairobi, Kenya, **3** Department of Medicine, Aga Khan University Hospital, Nairobi, Kenya, **4** Department of Pathology, Aga Khan University Hospital, Nairobi, Kenya

\* mansoor.saleh@aku.edu

**Data Availability Statement:** All relevant data are within the manuscript and its Supporting Information files.

## Abstract

### Background

Low dose radiation therapy (LDRT) has been used for non-malignant conditions since early 1900s based on the ability of single fractions between 50–150 cGy to inhibit cellular proliferation. Given scarcity of resources, poor access to vaccines and medical therapies within low and middle income countries, there is an urgent need to identify other cost-effective alternatives in management of COVID-19 pneumonia. We conducted a pilot phase Ib/II investigator-initiated clinical trial to assess the safety, feasibility, and toxicity of LDRT in patients with severe COVID-19 pneumonia at the Aga Khan University Hospital in Nairobi, Kenya. Additionally, we also assessed clinical benefit in terms of improvement in oxygenation at day 3 following LDRT and the ability to avoid mechanical ventilation at day 7 post LDRT.

### Methods

Patients with both polymerase chain reaction (PCR) and high-resolution computer tomogram (HRCT) confirmed severe COVID-19 pneumonia, not improving on conventional therapy including Dexamethasone and with increasing oxygen requirement were enrolled in the study. Patients on mechanical ventilation were excluded. Eligible patients received a single 100cGy fraction to the whole lung. In the absence of any dose limiting toxicity the study proposed to treat a total of 10 patients. The primary endpoints were to assess the safety/feasibility, and toxicity within the first 24 hours post LDRT. The secondary endpoints were to assess efficacy of LDRT at Day 3, 7, 14 and 28 post LDRT.

### Results

Ten patients were treated with LDRT. All (100%) of patients were able to complete LDRT without treatment related SAE within the first 24 hours post treatment. None of the patients

**Funding:** The author(s) received no specific funding for this work.

**Competing interests:** The authors have declared that no competing interests exist.

treated with LDRT experienced any acute toxicity as defined by change in clinical and respiratory status at 24hr following LDRT. Majority (90%) of patients avoided mechanical ventilation within 7 days of LDRT. Four patients (40%) demonstrated at least 25% improvement in oxygen requirements within 3 days. Six patients (60%) were discharged and remained off oxygen, whereas four progressed and died (1 due to sepsis and 3 in cytokine storm). Median time to discharge (n = 6) was 16.5 days and median time to death (n = 4) was 11.0 days. Patients who ultimately died showed elevated inflammatory markers including Ferritin, CRP and D-dimers as compared to those who were discharged alive.

## Conclusion

LDRT was feasible, safe and shows promise in the management of severe COVID-19 pneumonia including in patients progressing on conventional systemic treatment. Additional phase II trials are warranted to identify patients most likely to benefit from LDRT.

## Introduction

The natural history of COVID-19 pneumonia follows a predictable initial course consisting of initial infection followed by flu like symptoms accompanied by mild fever, body, and joint ache, altered taste and fatigue [1]. Symptoms peak around day 7 followed by a resurgence of immune response resulting in nearly 80% of patients recovering without major sequela [2]. Twenty percent of patients develop progressive shortness of breath and require oxygen therapy to keep pulse oxygen saturation > 94% [3], Remdesivir to halt viral replication [4] and Dexamethasone to ameliorate the enhanced inflammatory response that is felt to contribute to worsening lung function [5]. High-Resolution Computer Tomogram (HRCT) findings reveal patchy inflammation with ground glass pattern and intra alveolar edema. Increasing oxygen requirement is often followed by worsening pulmonary infiltrates and signs and symptoms of acute respiratory distress syndrome (ARDS) often leading to need for mechanical ventilation. The basis of this ARDS is felt to be related to an exuberant cellular inflammatory response lead by release of cytokines by pulmonary macrophages and immune effector cells, resulting in capillary leak syndrome, worsening gas exchange and progression of the patchy ground glass pattern on HRCT [6]. The global mortality rate of patients suffering progressive ARDS and cytokine storm can range between as 13–73% [7]. The anti-IL6R monoclonal antibody Tocilizumab has received regulatory approval and shown to be beneficial to avert intubation, improve survival and discharge from hospital in patients not on mechanical ventilation [8, 9]. Central to the worsening pulmonary picture appears to be an inflammatory response localized to the lung, accompanied by cytokine release leading to a systemic cytokine storm with fever, hypotension, and worsening gas exchange. Given this context it would seem relevant that treatment approaches targeted at inflammatory cells within the lung parenchyma would be useful in preventing the cascade of cytokine release and subsequent relentless progression into a cytokine storm and ARDS [10–12].

Despite initial skepticism, low dose radiation therapy (LDRT) has been successfully used to treat non-malignant conditions since early 1900s [13]. These have included bacterial and viral lobar and bronchopneumonia, as well as interstitial and atypical pneumonia [14]. Additional animal studies have supported these clinical findings and shown that low dose radiation treatment exerts an anti-inflammatory effect that leads to a rapid reversal of clinical symptoms,

facilitating disease resolution [15, 16]. The capacity of low doses of radiation to suppress inflammatory responses was most recently successfully reintroduced by the team from Emory University in the treatment of COVID-19 pneumonia [17]. Subsequently several centers have demonstrated comparable positive results depending on the timing of the LDRT intervention [18–21].

The pandemic has unmasked disparities between nations with inequitable distribution of proven therapies for the prevention and treatment of COVID-19 pneumonia. For this reason, Africa needs to develop cost effective and feasible alternatives for the treatment of COVID-19 pneumonia. Currently, access to Radiation therapy within LMIC(Low and Low Middle Income Countries) is limited, but with reliable electricity, increased placement of radiation therapy units, and capability to treat patients, LDRT could potentially as a "drug" one can make on site at low cost.

There is limited data currently predicting the optimal cohort that would most benefit from treatment with LDRT. The optimal timing for the institution of LDRT also remains unknown. Furthermore, inflammatory markers are often poor surrogates when used after the patients have been treated with multiple immunomodulatory drugs.

We conducted a pilot Ib/II investigator-initiated single center trial to assess the safety, feasibility, and toxicity of LDRT in patients with severe COVID-19 pneumonia. In addition, we assessed the improvement in oxygenation at day 3 following LDRT and the ability to avoid mechanical ventilation at day 7 post LDRT.

## Materials and methods

The study was approved by the Institutional Ethics and Review Committee at the Aga Khan University Hospital, Nairobi (AKUHN), a tertiary teaching and referral hospital in Kenya (#IERC/2020-111), as well as by the national health regulatory authorities. The trial was registered with the Pan African Clinical Trials Registry(PACTR202009769021840). All study patients were awake and alert and competent to provide consent as deemed by their referring MD. The patient had at least 24h to review the consent form prior to signing consent. All patients provided physical consent following a face to face visit by the investigator and in each case consent was verbally reconfirmed at the time of LDRT administration. An independent witness was present in the room at the time of consenting and this was documented by the witness signature on the consent form.

### Objectives

The primary objective of this study was to assess the feasibility, safety, and toxicity of a single dose of LDRT administered to patients with severe COVID-19 pneumonia. The secondary objectives were to determine improvement in oxygen requirement within 3 days following LDRT, ability to avoid mechanical ventilation within 7 days following LDRT and to determine time to discharge/death following LDRT.

### Cohort

Patients were eligible if they were age $\geq$ 18 years with PCR and HRCT confirmed COVID-19 pneumonia in conjunction with characteristic symptoms and need for oxygen to maintain a pulse oximeter saturation of $>$ 94%. Patients had to demonstrate clinical progression following conventional therapy available at the institution. Clinical progression was defined as inability to improve oxygenation despite current treatment, need to increase oxygenation and ventilatory support, and/or declining clinical status with standard of care management. Patients who received immunomodulatory therapy, including Tocilizumab, were only eligible if they had no

documented clinical improvement 72 hours after receiving the drug. Exclusion criteria included patients with pre-existing lung comorbidity such as severe COPD, severe uncontrolled asthma, heart failure or concomitant active systemic infection. Patients with pre-existing dependency on supplemental oxygen prior to diagnosis of COVID-19, Pregnant and/or planning to get pregnant within the next 6 months or hemodynamic instability that would preclude transfer to the radiation therapy suite were all excluded from the trial.

All patients received standard of care treatment with medications available within the institution and according to existing best practice at the time. This included oral or intravenous steroids (Dexamethasone/methylprednisolone), systemic antibiotics or Remdesivir. Patients with laboratory and clinical picture compatible with a cytokine storm syndrome received Tocilizumab as part of their treatment. All patients were encouraged to adopt a prone position to improve oxygenation.

## Treatment

Eligible patients received a single 100cGy fraction to the whole lung without shielding of the heart. Enrolled patients were transported on a stretcher with portable oxygen (max 15 L/min via nasal cannula or an FiO2 of 0.8 delivered via non-invasive ventilation or high flow oxygen therapy device) from the High Dependency Unit (HDU) or Intensive Care Unit (ICU) to the Radiation therapy (RT) suite using a dedicated patient hallway and elevator.

Where possible we used diagnostic CT scan DICOM images downloaded to our treatment planning system to calculate the monitor units/exposure for treatment. The dedicated CT simulator in the radiotherapy unit was only used if DICOM CT images were not available or unsuitable for calculations. When using the diagnostic CT images, the isocenter was established at the midpoint of the lung volumes. As all the treatments were planned using the Eclipse planning software (V.15), the software automatically corrected for lung density. The treatment was pre-planned to reduce the time spent in the unit. Static portal images were used to ensure proper coverage. The clavicle, as identified on the CT and clinically on the patient, was used as the reference set up point. For most patient the LDRT was administered in supine position, however, for patients rapidly de-saturating while supine we administered LDRT in prone position.

Study patients were treated after all regular patients had completed their treatments and strict institutional COVID-19 guidelines including sterilization of treatment room were observed. We instituted a 10-hour interval between the last COVID patient treated and first regular patient treated on the following day. All staff involved used personal protective equipment (PPE). Patients had their vital signs and pulse oxygen measured every 15 min for a total of 1h, then hourly for 4 hours following LDRT, and subsequently every 4h for the first 24h and then every 6h thereafter.

## Assessment

Serial laboratory tests, including complete blood counts (CBC), urea electrolytes & creatinine (UEC), liver function tests (LFT), D dimer, Ferritin and C-reactive protein (CRP) were performed on all patients prior to LDRT and on days 3, 7, 11, 14, 28 and at time of discharge. Disease severity was classified using the ordinal scores into: 1-discharged, 2-non-hospital ICU ward not requiring oxygen, 3-non-hospital ICU ward requiring oxygen, 4-ICU or non-ICU hospital ward, requiring non-invasive ventilation or high-flow oxygen, 5-ICU requiring intubation and mechanical ventilation, 6-ICU, requiring extracorporeal membrane oxygenation (ECMO) or mechanical ventilation and additional organ support (e.g., vasopressors, renal replacement therapy), 7-death. Ordinal scores were tracked on Day 1, 2, 3, 7, 14/28 and/or at

discharge. Serums samples collected were stored at -20 degrees Celsius for assessment of inflammatory cytokines at a future time point. The Phase Ib/II pilot study was designed to treat an initial 5 patients followed by review and assessment of feasibility, safety, and toxicity by the DSMC. In the absence of any dose limiting toxicity the study proposed to treat a total of 10 patients. Dose limiting toxicity was defined as a change in clinical and respiratory status within 24 hours post LDRT. Patients were followed for 28 days, or to discharge/death.

### Endpoints

Primary Endpoints included safety/feasibility, and toxicity following LDRT. Safety/feasibility was defined as percentage (%) of patients able to complete LDRT without treatment related SAE at 24h following LDRT. Toxicity was defined as the % patients without worsening of vital signs to assess clinical and respiratory status at 24h following LDRT (CTCAE acute toxicity criteria). Secondary Endpoints were defined as follows: % patient able to be weaned off pre-LDRT ventilatory or oxygen support at 3d post LDRT, % patients able to come off (or avoid) mechanical ventilation within 7 days following LDRT and % patients discharged or expired at 14d/28d post LDRT. Hematologic parameters were followed to determine any acute hematologic toxicity following LDRT

### Statistical analysis

Descriptive statistics was presented as frequencies and percentages for categorical data whereas medians and interquartile ranges for continuous data. All analyses were performed using the R statistical software and SPSS (IBM Version 20).

### Results

We enrolled a total of 10 patients on our study (Table 1). The median age was 59 years (range 42–72). Most of our patients were males (80%) and eighty percent (80%) of our patients were of African descent with 20% of Asian descent. Hypertension followed by diabetes was the commonest comorbidity. All patients had received Dexamethasone prior to study enrollment, 20% had received Remdesivir, and 70% had received Tocilizumab at least 72 hours prior to LDRT with no clinical improvement. Fig 1 shows the CONSORT flow.

The clinical parameters of the study patients at the time of LDRT are shown on Table 2.

Seventy percent (70%) of our patients were receiving non-invasive ventilation (NIV) at the time of enrollment with a fractionated oxygen requirement (FiO2) between 60–100%. All patients had a HRCT scan at the time of admission that revealed between 10–90% lung involvement. Median time from symptom onset to admission was 5 days (range: 2–13) and median to time from COVID-19 diagnosis to LDRT was 10 days (range: 5–20). Approximate transfer time from HDU/ICU to the RT suite and back was less than 30 min. Delivery of LDRT, including patient set-up took about 10 minutes. All patients received a single fraction

**Table 1. Baseline characteristics of patients enrolled in the study.**

| Patient | PT 1 | PT 2 | PT 3 | PT 4 | PT 5 | PT 6 | PT 7 | PT 8 | PT 9 | PT 10 |
|---|---|---|---|---|---|---|---|---|---|---|
| Age | 57 | 66 | 42 | 60 | 42 | 72 | 64 | 58 | 42 | 60 |
| Gender | M | F | M | M | M | M | M | F | M | M |
| Days from Onset of Symptoms to Admission | 13 | 4 | 2 | 5 | 4 | 6 | 6 | 5 | 4 | 8 |
| Days from Admission to LDRT | 23 | 10 | 10 | 7 | 4 | 8 | 7 | 13 | 6 | 15 |
| Days from symptom onset to LDRT | 36 | 14 | 12 | 12 | 8 | 14 | 13 | 18 | 10 | 23 |
| % Lung Involvement on HRCT At Admission | 80–90 | 70–80 | 50–60 | 30–40 | 50–60 | 10 | 10 | 50 | 40–50 | 50 |

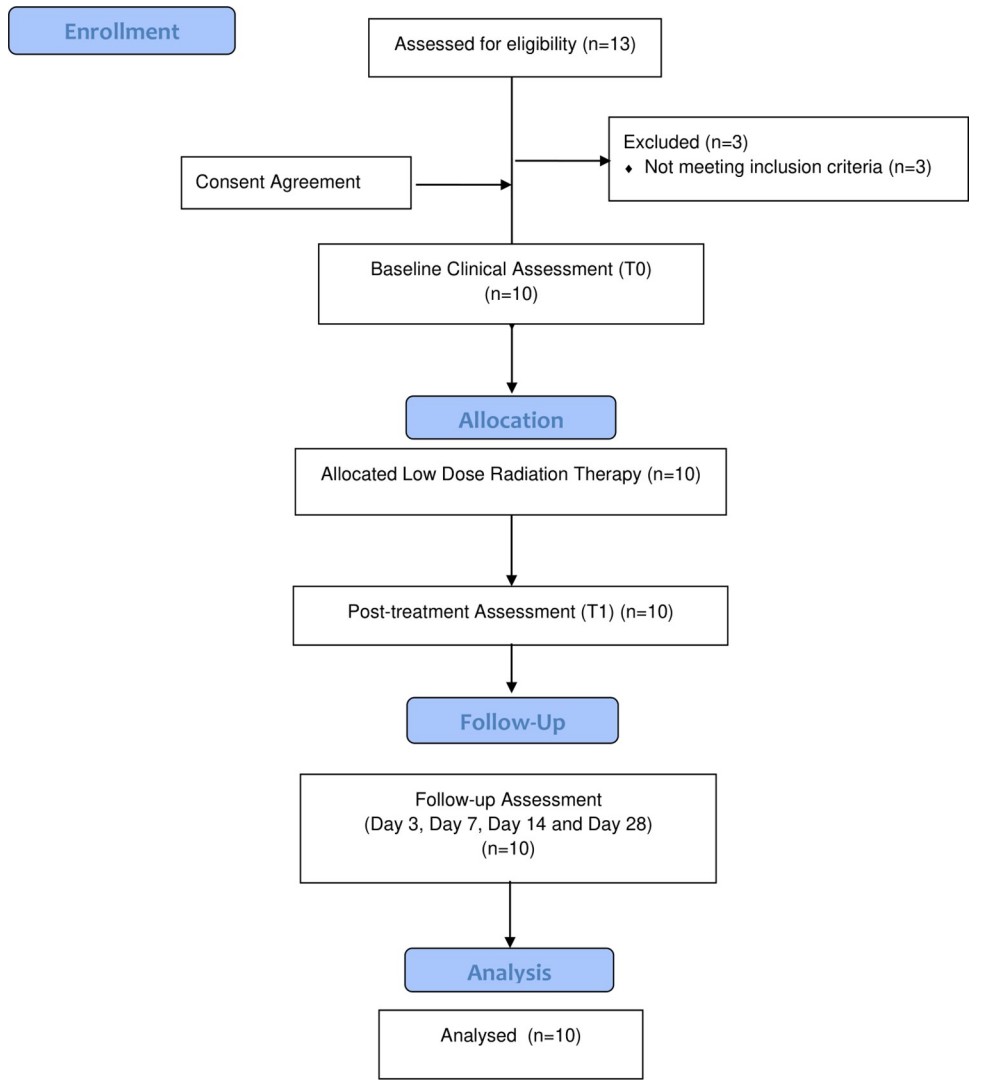

**Fig 1. Flow diagram of participants.**

of 100cGy delivered to the whole lung via opposing fields with no shielding of the cardiac silhouette

All patients tolerated LDRT with no appreciable change in the systolic & diastolic blood pressure, pulse rate, temperature, spO2 and respiratory rate during the four hours post-LDRT observation period.

All patients had stable clinical parameters following LDRT and 4 patients had improvement in oxygen requirement by day 3. Six patients had improvement in their oxygen parameters by day 7. Six of 10 patients enrolled on study were discharged home (3 patients were initially discharged on oxygen). Median time to discharge was 16.5 days (range: 4–28). All six patients who were ultimately discharged showed improved inflammatory markers (Ferritin and/or CRP) as well as d-dimers (Fig 3). Four patients died and median time to death was 11 days (range: 5–17) post LDRT. Death was due to sepsis (n = 1) or worsening COVID-19 and ARDS (n = 3). Three of the four patients were intubated on day 9, 8 and 11 post LDRT respectively. The fourth patient declined resuscitation measures. All 4 patients who ultimately died

**Table 2. Patient clinical parameters at the time of LDRT.**

| Patient Study Number | PT 1 | PT 2 | PT 3 | PT 4 | PT 5 | PT 6 | PT 7 | PT 8 | PT 9 | PT 10 |
|---|---|---|---|---|---|---|---|---|---|---|
| Ordinal score at time of LDRT | 4 | 4 | 4 | 4 | 4 | 4 | 3 | 3 | 4 | 4 |
| Therapy prior to LDRT: | | | | | | | | | | |
| a. Oxygen | Y | Y | Y | Y | Y | Y | Y | Y | Y | Y |
| b. Steroids | Y | Y | Y | Y | Y | Y | Y | Y | Y | Y |
| c. Remdesivir | N | N | N | N | N | N | N | N | Y | Y |
| d. Tocilizumab | Y | Y | Y | Y | N | N | N | Y | Y | Y |
| Oxygen parameters at time of LDRT: | HFNC FIO2-0.7 | NIV FIO2-0.9 | NRM 15L | HFNC FIO2-1.0 | NIV FIO2-1.0 | FM 10L | HFNC FIO2-1.0 | FM 5L | NIV FIO2-1.0 | NIV FIO2-0.6 |
| Outcome | A | A | A | A | A | D | D | D | A | D |

Abbreviations: *Y: Yes; N: No; HFNC: High Flow Nasal Cannula; NIV: Non-Invasive Ventilation; NRM: Non-Rebreather Mask; FM: Face Mask. Ordinal score: 1: discharged, 2: non-hospital ICU ward not requiring oxygen, 3: non-hospital ICU ward requiring oxygen, 4: ICU/non-ICU requiring NIV or high flow nasal cannula, 5: ICU requiring intubation and mechanical ventilation, 6: ICU requiring ECMO/organ support, 7: death; FiO2: fractionated oxygen requirement. A-alive, D-Dead*

experienced worsening inflammatory markers. The CRP, ferritin and D-dimers for all patients from treatment to discharge or death are depicted in Fig 2. Fig 3 (spaghetti plot) shows the values for the individual patients grouped into alive/discharge or died. There appears to be a trend of a higher baseline and an upward trajectory in CRP, D-dimer and Ferritin for the patients who died as compared to the patients who were alive at the end of the study. The same data provided as a box plot is included as a S1 Fig. Hematological indices were monitored regularly post enrolment and revealed no acute toxicity post treatment (Fig 2).

## Discussion

Investigators from Emory University were the first to study the safety of LDRT in COVID-19 pneumonia [17]. Their cohort of 10, primarily elderly COVID-19 PCR positive patients on

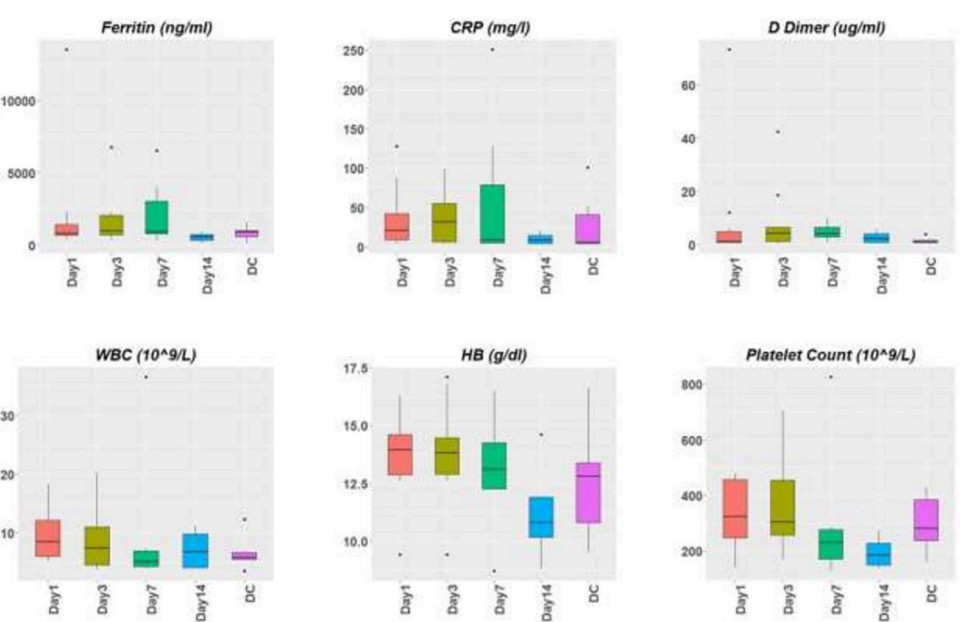

**Fig 2. Inflammatory and Hematological parameters for all patients from enrolment to discharge/death.**

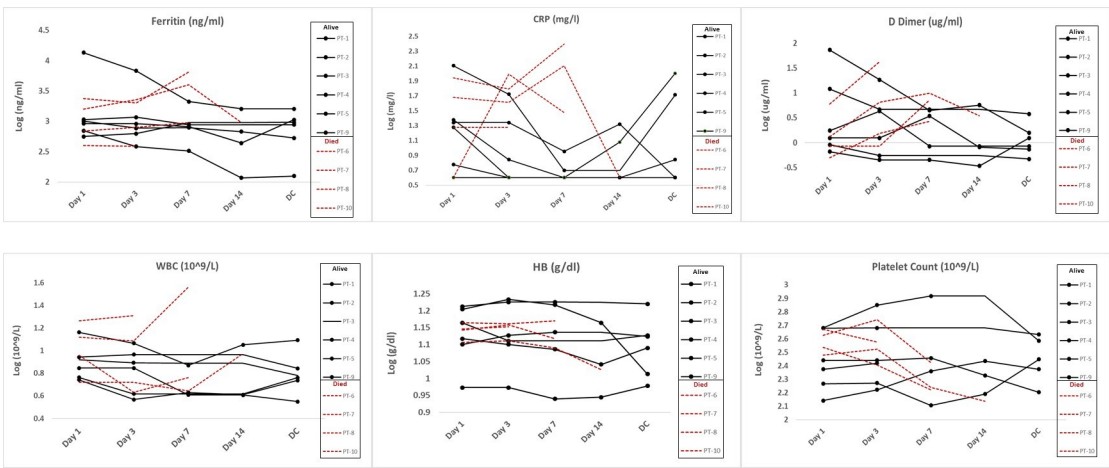

**Fig 3. Spaghetti plots depicting inflammatory and Hematologic parameters for all patients from admission to discharge/death, comparing those who lived versus those who died.**

oxygen per nasal cannula (1.5-6l/min) were treated early in their clinical course (median of 4.5 days' post hospitalization) using whole lung doses of 1.5 Gy. Compared to historical age matched control patients treated with LDRT demonstrated a significant improvement in the time to recovery, time to discharge and intubation free rates. In December 2021, the same investigators published data from a phase II trial which enrolled 20 patients receiving Dexamethasone and/or Remdesivir to receive a single dose of 1.5Gy. This cohort included younger patients (median age 64.5 years), median time to LDRT was 3 days and the patients were excluded if they required more than 15L of oxygen. They reported a reduction in intubation rates from 32% to 14%, 80% survival of patients treated with LDRT, reduction in time to clinical recovery and time to discharge compared to age matched controls [18]. In contrast, the patients in our study were younger, had higher oxygen requirements with majority being on noninvasive positive pressure ventilation, received LDRT late in their hospitalization (median of 10 days post hospitalization), and a majority had received systemic therapy including Tocilizumab.

Since the initial experience at Emory University, several investigators have conducted similar studies on a diverse set of COVID-19 patients using varying doses of LDRT (Table 3). Sharma et al from the All India Institute of Medical Sciences (AIIMS) enrolled 10 patients

**Table 3. Summary table with select publications investigating role of LDRT in COVID-19 management.**

| PUBLICATION | N | MED. AGE | LDRT DOSE | TIME TO LDRT | PRIOR THERAPY | OUTCOME |
|---|---|---|---|---|---|---|
| Hess et al 2020 [17] | 5 | 90 | 1.5Gy | 5 days | Azithromycin | • Alive: 4<br>• Mechanically ventilated: 1 |
| Hess et al 2021[18] | 20 | 64.5 | 1.5Gy | 3 days | Dexamethasone | • Alive: 16<br>• Dead: 4 |
| | | | | | Remdesivir | |
| Sharma et al [19] | 10 | 51 | 70 cGy | 3 days | Steroids | • Alive: 9<br>• Dead: 1 |
| Ameri et al [20] | 10 | 75 | 0.5Gy/1.0Gy | 2–4 days | None | • Alive: 5<br>• Dead: 5 |
| Sanmamed et al [21] | 9 | 66 | 100cGy | 52 days | Steroids | • Alive: 8<br>• Dead: 2 |
| | | | | | Tocilizumab (n = 3) | |
| | | | | | Remdesivir (n = 1) | |

(mean: 51 years) with moderate to severe COVID-19 within a median of 3 days following hospital admission and received 70cGy to both lungs [19]. They found a response/ recovery rate of 90% with no evidence of acute toxicity. Hess et al and Ameri et al similarly enrolled 5 patients each and administered LDRT at 1.5Gy and 0.5Gy respectively [17, 20]. Patients were enrolled within 5 days of admission and achieved a response rate approaching 80%. Sanmamed et al had a similar cohort to our study with younger patients but treated much later in their disease course at a median time from admission of 52 days with 30% of their patients having received Tocilizumab. They demonstrated that the 8 surviving patients had declining inflammatory markers (CRP, D-dimers, ferritin) within a week following LDRT [21]. Most of these studies were conducted prior to the broad use of systemic anti-viral therapy (Remdesivir) or anti-inflammatory treatment (Tocilizumab). Only 30% of their patients were discharged home still requiring oxygen likely reflecting a degree of irreversible lung damage following prolong COVID-19 pneumonia associated hospitalization.

Our study is the first to be conducted in sub-Saharan Africa. In Kenya, a population of over 47 million has access to a total of only 12 radiation therapy units across 47 counties [22–24]. Our primary objective was to assess the feasibility to safely administer LDRT within a tertiary care university center at a time when majority of the in-patient beds were occupied by patients with severe COVID-19 pneumonia. Transport and treatment from the hospital bed to the radiation therapy unit and back took less than 30 min and all patients tolerated the treatment well with no alteration of clinical parameter over the 24h following LDRT. Our patients were younger (median: 59 years) with a comorbidity profile similar to patients in other trials. However, compared to other trials, our patients were sicker with high oxygen requirement and enrolled after having failed protocol mandated standard of care treatment which included Tocilizumab in 7 of the 10 patients. Patient treated with Tocilizumab had to be observed for lack of response for at least 72h prior to being considered for LDRT, thus further delaying onset of LDRT. While no radiographic imaging was performed prior to LDRT, all our patients demonstrated clinical signs of extensive lung involvement associated with high oxygen requirements necessitating use of non-invasive ventilation [17, 19, 20].

Of the 6 patients who were discharged home (median time 16.5 days (range:4–28), three went home off oxygen while 3 required home oxygen for up to 1–2 months post discharge. We noted a downward trend in inflammatory markers in these patients as compared to patients who died. However, the small sample size and incomplete data points for some patients weakens this observation. Of the 4 patients who died, 2 were Tocilizumab naive but developed cytokine storm at 5 and 7 days post LDRT respectively, necessitating administration of Tocilizumab and subsequently required intubation. Cause of death for the 4 patients (median time to death 11.0 days (range:5–17) was COVID-19 progression for three patients and bacterial sepsis for one patient. Our study ably demonstrated the feasibility and safety of LDRT in the treatment of severe COVID-19 pneumonia in a low resource setting. None of the patients enrolled in our study experienced any dose related acute toxicity. While long term toxicity was beyond the scope this trial none of our living patients have reported pulmonary or hematologic toxicity.

The cost of a single fraction of LDRT at our institution is USD 30 and compares very favorable to the average cost of Tocilizumab at USD 1,300 for a 400 mg dose. The cost for Remdesivir is approx. USD1000 for a 5-day treatment. The average daily charge for an ICU bed for an intubated patient in Kenya is approx. USD 699 [25] and this does not take into account the additional charges for high intensity clinical care of a COVID-19 patient. While efficacy was not the primary endpoint, 60% of our patients were discharged and avoided intubation during a critical phase of their illness. Cost constraints limited out ability to perform HRCT prior to LDRT to determine extent of lung involvement at the time of LDRT. In the limited sample of

PAO2/FIO2 (P/F) ratio, we did observe a decline in P/F in patients who did not survive, and this accompanied the worsening inflammatory markers. Whether P/F and inflammatory markers would be predictive parameter to determine selection of patients and timing of LDRT remains to be determined. We were unable to identify a matched control group since patients not enrolled on this trial had incomplete documentation and limited laboratory tests during their hospitalization. In the absence of universal health coverage standard of care is often dependent on affordability.

In the context of a low resource setting where the patient often bears most of the cost of care, the question whether LDRT could be instituted early and would be more cost efficient as opposed to delayed after having exhausted all, often more expensive, conventional options remains to be addressed and will require well designed clinical trials which should include pharmaco-economic endpoints. Future waves of COVID-19 infection may well provide the platform to address these critical questions from the clinical, ethical, and socio-economic standpoint. Randomized studies are currently ongoing to rigorously determine the effectiveness of LDRT. These results if conclusive may allow an objective discussion regarding the early administration of LDRT in a low resource setting where such capability exists. Unfortunately, most sub-Saharan nations do not have the luxury of having sufficient radiation therapy facilities even to meet the needs of their cancer patients. However, if future phase II/III provide evidence for the cost effectiveness of radiation therapy as a treatment for COVID-19 pneumonia, this may add impetuous for low resource nations to see the broader utility of radiation therapy units.

## Conclusion

Low dose radiation therapy is a feasible option with no acute toxicity in the management of severe COVID-19 in the setting of a low resource country especially at institutions that have the capability to deliver radiation therapy. Our findings are comparable to what has been observed by other investigators and demonstrates the utility of LDRT in this setting. Given that COVID-19 has yet to be eradicated, and with ongoing new waves of severe infection predicted, the issue of employing LDRT early in the clinical management of severe COVID-19 pneumonia needs to be studied further, especially in a setting where the availability of expensive immune modulating agents is limited or unaffordable.

## Supporting information

**S1 Checklist.**
(DOCX)

**S1 Fig. Comparison of the inflammatory and hematological parameters between patients that died versus those that survived post LDRT.**
(TIF)

**S1 Protocol.**
(DOCX)

**S1 Data.**
(CSV)

## Acknowledgments

We would like to gratefully acknowledge the assistance of our nursing staff who managed the COVID-19 patients and contributed to the execution of this study, and to our patients and their families for agreeing to participate in this clinical trial.

## Author Contributions

**Conceptualization:** Mansoor Saleh, Farrok Karsan, Angela Waweru, Martin Musumbi, Reena Shah.

**Data curation:** Jasmit Shah, Heldah Amariati.

**Formal analysis:** Karishma Sharma, Jasmit Shah.

**Funding acquisition:** Mansoor Saleh.

**Investigation:** Mansoor Saleh, Karishma Sharma, Farrok Karsan, Angela Waweru, Martin Musumbi, Reena Shah, Shahin Sayed, Innocent Abayo, Noureen Karimi, Sehrish Rupani, Grace Kirathe.

**Methodology:** Mansoor Saleh, Jasmit Shah, Farrok Karsan, Angela Waweru, Martin Musumbi, Reena Shah, Shahin Sayed, Innocent Abayo, Noureen Karimi, Stacey Gondi.

**Project administration:** Mansoor Saleh, Karishma Sharma, Farrok Karsan, Angela Waweru, Martin Musumbi, Shahin Sayed, Innocent Abayo, Noureen Karimi, Stacey Gondi, Sehrish Rupani, Grace Kirathe, Heldah Amariati.

**Resources:** Mansoor Saleh, Martin Musumbi, Innocent Abayo, Noureen Karimi.

**Software:** Jasmit Shah.

**Supervision:** Mansoor Saleh, Karishma Sharma, Farrok Karsan, Angela Waweru, Martin Musumbi, Reena Shah, Shahin Sayed, Innocent Abayo, Noureen Karimi, Stacey Gondi, Grace Kirathe.

**Validation:** Jasmit Shah.

**Visualization:** Jasmit Shah.

**Writing – original draft:** Mansoor Saleh, Karishma Sharma, Jasmit Shah, Farrok Karsan, Angela Waweru, Martin Musumbi, Reena Shah, Shahin Sayed, Innocent Abayo.

**Writing – review & editing:** Mansoor Saleh, Karishma Sharma, Jasmit Shah, Farrok Karsan, Angela Waweru, Martin Musumbi, Reena Shah, Shahin Sayed, Innocent Abayo, Noureen Karimi, Stacey Gondi, Sehrish Rupani, Grace Kirathe, Heldah Amariati.

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
