## [Decision Letter · Decision Letter 0]

10 Jan 2022

PONE-D-21-35607A Pilot Phase Ib/II Study of Whole-Lung Low Dose Radiation Therapy (LDRT) For the Treatment of Severe COVID-19 Pneumonia: First Experience from AfricaPLOS ONE

Dear Dr. Saleh,

Thank you for submitting your manuscript to PLOS ONE. After careful consideration, we feel that it has merit but does not fully meet PLOS ONE’s publication criteria as it currently stands. Therefore, we invite you to submit a revised version of the manuscript that addresses the points raised during the review process.

The reviewers have identified multiple points that warrant clarification. If you chose to resubmit to this journal please provide a point-by-point report of the edits and changes made in response to reviewer comments.

We look forward to receiving your revised manuscript.

Kind regards,

Randall J. Kimple

Academic Editor

PLOS ONE

Journal Requirements:

2. Please describe in your methods section how capacity to provide consent was determined for the participants in this study. Please also state whether your ethics committee or IRB approved this consent procedure. If you did not assess capacity to consent please briefly outline why this was not necessary in this case.

Additional Editor Comments:

This manuscript has been reviewed by two individuals with expertise in the field. They have identified multiple points to clarify in order to improve the quality of the work.

Reviewers' comments:

Reviewer's Responses to Questions

**Comments to the Author**

1. Is the manuscript technically sound, and do the data support the conclusions?

Reviewer #1: No

Reviewer #2: Yes

2. Has the statistical analysis been performed appropriately and rigorously? 

Reviewer #1: No

Reviewer #2: Yes

3. Have the authors made all data underlying the findings in their manuscript fully available?

Reviewer #1: No

Reviewer #2: Yes

4. Is the manuscript presented in an intelligible fashion and written in standard English?

Reviewer #1: Yes

Reviewer #2: Yes

5. Review Comments to the Author

Reviewer #1: The authors report use of radiation therapy in a phase IB/II as a novel treatment for patients hospitalized with COVID-19 pneumonia. This controversial topic is important because it challenges the current perspective on the use of radiotherapy primarily for neoplastic disease. However, it needs to be reported in a way to adequately address concern and skepticism that ionizing radiation may be ineffective in treating viral pulmonary infection and also may result in acute or late toxicity.

This clinical trial combines assessing the risks of clinical toxicity (phase IB) with the potential to assess clinical benefit (phase II) and has IRB approval. The authors capture a great deal of important clinical information in this report, but it may be helpful to provide more detail in some areas.

Also, many investigators at elite institutions in non-LMICs may not appreciate the value of less conventional treatments given scarcity and poor access to vaccines and expensive medical therapies. It may be worth highlighting this point more in the introduction and discussion.

Given the small size of this trial, most of what we can interpret would be related to acute treatment toxicity. Unlike the Emory study, which reported data compared to similar patients not receiving radiation, this trial can give some sense of how patients fared after LDRT but there will be selection biases on who is included in the trial.

For the abstract:

1. P. 9, line 29: Not sure the remdesevir/tocilizumab information is needed, unless there is room. That doesn’t define the cohort included. Consider discussing tocilizumab use and number of patients in cytokine storm at time of LDRT in Results.

2. P. 9, lines 31-3: Exclude, would favor more information on how safety and clinical endpoints were defined and reported.

3. P. 9 Will discuss below but for results start with defining key aspects of cohort, and it should include reporting some metrics of its acute safety defined in manuscript’s Methods.

4. P.10, lines 44-47: Consider moving the discussion of LMIC limited resources to Introduction, and limit conclusions to acute toxicity, confirm other phase I trials and indicating need for larger phase II/III trials

For the body of the manuscript:

Introduction

The authors correctly discuss some of the pathophysiology of COVID-19 pneumonia and the need for more effective therapies. But there is a reason many of the early reported trials are in LMICs – the pandemic has worsened disparities and even ‘proven’ vaccine and therapies are not shared equitably on a global scale. So alternative strategies may need to be developed.

Radiation therapy may be limited access in LMICs but with reliable electricity and capacity to treat patients, LDRT is a “drug” you can make on site at low cost. The authors may want to emphasize this point a bit more as the value proposition and a rationale for clinical trials that might not seem necessary to some.

Caveat on this point will be that p.12 lines 83-87 capture a limitation in many African nations – poor access to radiotherapy for cancer care already. That is a limitation currently but also an opportunity to address in the Discussion as a potential future rationale for infrastructure investment in radiation facilities that may have value for cancer but also for infection, if phase III trials ultimately demonstrate clinical benefit.

One of the considerations in discussing LDRT for patients cytokine storm is that the anti-inflammatory effects has been hypothesized to work best before the cytokine storm takes place. Because the clinical trial permitted inclusion of patients already in the storm phase receiving tocilizumab, some may perceive this factor as a relative weakness of the trial. Consider framing it in a way you can touch up on it in the Introduction that fits with reporting it and then recognizing it as a potential weakness in the trial. Many are still trying to define if it works and an optimal cohort that benefits.

Methods

1. p. 13 lines 94-103: Discuss trial approval and eligibility (lines 99-103) first, along with exclusion criteria. Was the inclusion permitted based upon no clinical improvement (defined how?) or clinical progression despite dexamethasone and oxygen?Anticipate that many in the clinical oncology/radiation oncology community may have concerns with the inclusion of young patients that may have less need for LDRT (more likely to recover) and more risk of late effects not captured in a phase IB trial. Were there other exclusion criteria other than not on mechanical ventilation (e.g. prior thoracic radiation, prior chemotherapy, others)?

2. It may be helpful to have subsections in italics (or separate paragraphs) for Cohort, Patient Assessment, Treatment (describing the LDRT), defining endpoints and then statistical/analytic methods.

3. P.14, line 114: Some more detail, even if brief, would be helpful. Planned in CT simulator, clinical setup on linear accelerator? Prescribed to isocenter, with or without heterogeneity corrections? Part of assessing feasibility includes the ability to deliver the treatment, so the authors should provide more information for the reader to assess the difficulty of accomplishing the protocol therapy. Lines 128-135 should be with line 114.

4. The authors should provide more detail on safety monitoring to meet the stated primary objective. How do you define/report feasibility? Hematologic toxicity is probably the primary immediate safety risk, given the large amount of bone marrow irradiated and likely including a lot of the spleen. Other trials have reported baseline and trends in hematologic indices, which would be helpful in this case. The authors focus more on vital signs, but presumably hematologic indices can also be reported since they were assessed (line 118).

5. Secondary objectives are reasonable, were these based upon other trials or developed prior to publication of other reports? It’s all a rapidly developing area, but it deserves some clarity here.

Results

1. P. 15, line 146: For a small trial, IQR may be less relevant than full range, especially given some concerns about younger patients treated. Consider reporting total range, not IQR.

2. P.16, line 156 and other tables: label each column by patient (PT) not LD for low dose treatment.

3. P.16, line 156: Table 1 reports performance status at admission as 0. What scale is used, and if admitted presumably ill enough to not be a 0? Consider omitting.

4. P.16, line 156: Table 1 should report time from symptoms to LDRT. Consider moving performance status down in the table and including all temporal related metrics together.

5. Baseline clinical parameters should include hematologic data.

6. P.17, line 162 Table 2: no description of ordinal score in text to permit interpretation.

7. P. 18, lines 172-174 should be with the following paragraph, bringing all the radiation details together.

8. P.19, line 181: Table 3 – Need to include hematologic parameters as priority over inflammatory markers to demonstrate safety first. And consider reporting if toxicity recorded as yes/no or toxicity grade, rather than reporting numeric hematologic indices

9. P.19, lines 181: Table 3 – what is the difference between “–“ and “n/a” – did some labs not get drawn? Patient 3 (LD3) has three empty cells. Needs clearer organization for the reader to interpret.

10. P.19, line 181: Table 3 – patient 10 is listed too early, biased by end outcome (death). Patients should be presented similarly through all Tables.

11. For the inflammatory markers of interest, consider graphing it in a figure, rather than tabular format.

Discussion

1. P. 20, line 199: the Emory study was still essentially still a phase I, so didn’t evaluate effectiveness as much as safety.

2. The authors’ trial included a lot of patients with symptoms for a fairly long time and who required tocilizumab. It may be that the timing of intervention was too late in the pathophysiology of evolving COVID pneumonia. Despite the interest in LDRT, a randomized trial of intubated patients showed no clinical benefit. The authors need to put their trial cohort (younger, more advanced disease than Emory) on spectrum of severity compared to other trials. Part of the learning from these early trials will be defining an optimal cohort for phase II or III trials.

3. The authors should emphasize the safety of LDRT for acute effects, after clearly presenting them in Results. There also needs to be some acknowledgement of late effects as something beyond the scope of this trial that should be included in safety monitoring in future clinical trials.

4. Making the cost argument is valid. Given that treating patients once tocilizumab is needed for cytokine storm, should LDRT be considered earlier in the disease process for trials? It could result in cost savings if LDRT works. If not, why not?

5. P.24, lines 270-272: This point on scarcity of linear accelerators needs to fit coherently from Introduction to Discussion. An opportunity to highlight need for investment to help cancer patients regardless of LDRT trial findings, made stronger if future studies of LDRT show a benefit.

Conclusion

1. Emphasize feasibility and acute safety.

2. Recovery rate is can’t be attributed to LDRT success in line 277.

3. Lines 281-2: recommend being genetic as ‘drug therapy’ rather than specifying immune modulating agents. Other effective drug classes may also be too expensive.

4. Lines 282-3: Final sentence not necessary

Reviewer #2: The authors present the fist study from African content on the use of LD-RT for COVID-19 ARDS. The study adds significant value to the current, limited understanding of the role of LD-RT for COVID-19 ARDS, and it's potential reproducibility of signal across other institutions across the world. Reproducibility of the signal across the world, in a different cohort of patients from Africa adds significant value, and is scientifically a notable accomplishment.

The authors also do a good job on only doing a small, feasibility study of 10 patients and asking the first level, simple safety and potential efficacy signal. Their patient selection of advanced, oxygen dependent patients is also reasonable. They chose patients that were not responding to other therapies and offered them the option of LD-RT and appears to find some clinical benefit. Their results of 6 patients improving is also in line with findings from others. There appears to be a correlation between inflammatory reductions post LD-RT and those patients that improve.

Mino edits:

1) Line 216, page 21. Did you mean to say 100 cGy instead of "gGy"?

2) Can the authors discuss preLD-RT rises in CRP, Ferritin, and other biomarkers and compare them with post LD-RT changes? In the work by Hess et al, they saw signficant rising trends preLD-RT, and then preciptiious drops in some of these post LD-RT. Of these biomarkers, CRP appeared to be the most correlated. Did the authors see this in their own data sets (comparing pre/post LD-RT biomarker changes).. See Hess et al. Radiotherapy and Oncology. Vol 165. December 2021. Pages 20-31.

https://www.sciencedirect.com/science/article/pii/S0167814021087594?dgcid=rss_sd_all

3) The authors can also use the reference by Hess et al to add the comment that LD-RT could potentially have independently benfits patients, despite having gotten Remedesivir and Steroids.

4)The authors can add the following preclinical studies to also support their notion that LD-RT may potentiall help COVID-19 ARDS : (Meziani et al. Int. J. Radiation Oncology Biology and Physicis. Volume 110. Issue 5. P 1283-1294. August 01, 2021). This is preclinical study looking at LD-RT in mouse using three different ARDS models and demonstarted that LD-RT helped convert inflammatory environment into an anti-inflammatory environment.

Second paper to considering adding: Jackson et al. Low Dose Radiotherapy forCOVID-19 Lung Disease: Preclincical Efficay in Bleomycin Model"... 2022. Jan 1; 112(1): 197-211. Int J. Radiation Oncology Biol & Physics. https://pubmed.ncbi.nlm.nih.gov/34478832/

6. PLOS authors have the option to publish the peer review history of their article (what does this mean?). If published, this will include your full peer review and any attached files.

Reviewer #1: **Yes: **Matthew Katz, MD

Reviewer #2: No

---

## [Author Response · Author response to Decision Letter 0]

1 Feb 2022

A Pilot Phase Ib/II Study of Whole-Lung Low Dose Radiation Therapy (LDRT) For the Treatment of Severe COVID-19 Pneumonia: First Experience from Africa

PLOS ONE: Reviewer comments with responses

Reviewer #1

The authors report use of radiation therapy in a phase IB/II as a novel treatment for patients hospitalized with COVID-19 pneumonia. This controversial topic is important because it challenges the current perspective on the use of radiotherapy primarily for neoplastic disease. However, it needs to be reported in a way to adequately address concern and skepticism that ionizing radiation may be ineffective in treating viral pulmonary infection and also may result in acute or late toxicity.

Response:

The reviewers point is well taken since there remains skepticism about the anti-inflammatory and immune modulating role of LDRT and its relevance as a “therapeutic agent” in the treatment of COVID 19 pneumonia. Our work, together with that of others demonstrates the feasibility, lack of acute toxicity and potential benefit of LDRT in this context. We have revised various aspect within our manuscript to reflect the reviewers input.

This clinical trial combines assessing the risks of clinical toxicity (phase IB) with the potential to assess clinical benefit (phase II) and has IRB approval. The authors capture a great deal of important clinical information in this report, but it may be helpful to provide more detail in some areas.

Also, many investigators at elite institutions in non-LMICs may not appreciate the value of less conventional treatments given scarcity and poor access to vaccines and expensive medical therapies. It may be worth highlighting this point more in the introduction and discussion.

Response:

The reviewer’s insight and input is much appreciated and we have revised our manuscript and added some details to incorporate this suggestion. 

Refer to P. 5 Line 91-96

Given the small size of this trial, most of what we can interpret would be related to acute treatment toxicity. Unlike the Emory study, which reported data compared to similar patients not receiving radiation, this trial can give some sense of how patients fared after LDRT but there will be selection biases on who is included in the trial.

Response:

As pointed out in our manuscript, our selection criteria were much more stringent than in the Emory study especially since our IRB required that all conventional therapy, including Tocilizumab, be offered and only those patients not improving be eligible to receive LDRT. Unfortunately, in the absence of an electronic medical record and lack of universal health coverage, patients not on study often have incomplete documentation and limited laboratory and diagnostic tests performed. Consequently, it has not been possible to identify an accurately matched control group as was done by the colleagues at Emory. We have added this as a limitation in our discussion. 

Please Refer to Page 19 Lines 327-330

For the abstract:

1. P. 9, line 29: Not sure the remdesevir/tocilizumab information is needed, unless there is room. That doesn’t define the cohort included. Consider discussing tocilizumab use and number of patients in cytokine storm at time of LDRT in Results.

Response:

The authors accept this suggestion by the reviewers and have omitted the statement from the abstract.

2. P. 9, lines 31-3: Exclude, would favor more information on how safety and clinical endpoints were defined and reported.

Response:

The authors accept this suggestion by the reviewer and have omitted information relating to IRB approvals. Safety endpoint were defined as acute toxicity within 24h of LDRT and have been outlined in the abstract and methods section.

Refer to P. 2 Line 36-38 & P. 9 Line 181-189

3. P. 9 Will discuss below but for results start with defining key aspects of cohort, and it should include reporting some metrics of its acute safety defined in manuscript’s Methods.

Response:

Our revised manuscript reflects the suggestion made the reviewer. We have included metrics of acute safety from LDRT at the beginning of the Results section in the Abstract. 

Refer to P. 3 Line 41-44

4. P.10, lines 44-47: Consider moving the discussion of LMIC limited resources to Introduction, and limit conclusions to acute toxicity, confirm other phase I trials and indicating need for larger phase II/III trials

Response:

We have revised our Discussion as suggested by the reviewer to include the issue of limited resources within the LMIC, highlighted (lack of) acute toxicity and alluded to the need for additional clinical trials to determine the optimal role and timing of LDRT in the treatment of COVID 19 pneumonia. We have also provided additional data (Fig 2 and 3) to demonstrate the importance of inflammatory markers to potentially predict outcome and, underscore the lack of acute hematologic toxicity. We have moved information regarding scarcity of resources and poor access to medical therapies in LMIC to the Background and Introduction. We have also included a statement on the need for further clinical trials to identify patients that are most likely to benefit from LDRT. 

Refer to P. 2/3 Line 21-23 & 54-55. 

For the body of the manuscript:

Introduction

The authors correctly discuss some of the pathophysiology of COVID-19 pneumonia and the need for more effective therapies. But there is a reason many of the early reported trials are in LMICs – the pandemic has worsened disparities and even ‘proven’ vaccine and therapies are not shared equitably on a global scale. So alternative strategies may need to be developed.

Radiation therapy may be limited access in LMICs but with reliable electricity and capacity to treat patients, LDRT is a “drug” you can make on site at low cost. The authors may want to emphasize this point a bit more as the value proposition and a rationale for clinical trials that might not seem necessary to some.

Response:

The reviewer’s point is well taken and we have included relevant commentary in our Discussion and also revised our Introduction. 

Refer to P. 5 lines 91-96

Caveat on this point will be that p.12 lines 83-87 capture a limitation in many African nations – poor access to radiotherapy for cancer care already. That is a limitation currently but also an opportunity to address in the Discussion as a potential future rationale for infrastructure investment in radiation facilities that may have value for cancer but also for infection, if phase III trials ultimately demonstrate clinical benefit.

Response:

The use of LDRT in COVID 19 pneumonia has not been explored in the LMIC. Ours is the first publication to study this modality, which as the reviewer points out may serve as a very affordable modality if more radiation therapy units were available in the LMIC. Sub Saharan Africa only has approx. 200 linear accelerators for a patient population of nearly 1 billion across 54 countries. We hope that peer reviewed publications from the LMIC like ours will demonstrate the value of radiation therapy units beyond they anti-cancer role and consider the capability of radiation therapy units to deliver LDRT, especially in the new normal environment of ongoing Covid 19 surges that have significantly impacted the LMIC. If ongoing trials confirm the effectiveness of LDRT, a single dose of LDRT would be the most cost effective modality, after steroids and oxygen, for the treatment of Covid 19.

Refer to P. 20 Line 332-340

One of the considerations in discussing LDRT for patients cytokine storm is that the anti-inflammatory effects has been hypothesized to work best before the cytokine storm takes place. Because the clinical trial permitted inclusion of patients already in the storm phase receiving tocilizumab, some may perceive this factor as a relative weakness of the trial. Consider framing it in a way you can touch up on it in the Introduction that fits with reporting it and then recognizing it as a potential weakness in the trial. Many are still trying to define if it works and an optimal cohort that benefits.

Response:

The reviewer’s point is well taken in that the treatment to counter the cytokine related pulmonary toxicity is best applied in the early setting of the cytokine storm. Our IRB felt it would be unethical to administer experimental LDRT ahead of FDA approved agents specifically aimed at combating the effects of systemic cytokines e.g. Tocilizumab. Our study was thus to allow patients to receive Tocilizumab and only enroll onto the LDRT study if they demonstrated no benefit. We are hopeful that our manuscript demonstrating the feasibility and lack of acute toxicity would support revisiting the role of LDRT for future COVID 19 waves especially in the LMIC. Various clinical trial strategies could be considered to test early administration of LDRT e.g. Tocilizumab +/- LDRT in patients showing early signs of cytokine storm, or compassionate use of LDRT early following failure of steroid therapy. Future trials may well receive support (or otherwise) from ongoing clinical trials referenced in our manuscript.

Refer to P. 5&6 lines 98-101

Methods

1. p. 13 lines 94-103: Discuss trial approval and eligibility (lines 99-103) first, along with exclusion criteria. Was the inclusion permitted based upon no clinical improvement (defined how?) or clinical progression despite dexamethasone and oxygen?

Response:

Our Method section has been revised to clarify the eligibility criteria

Anticipate that many in the clinical oncology/radiation oncology community may have concerns with the inclusion of young patients that may have less need for LDRT (more likely to recover) and more risk of late effects not captured in a phase IB trial. Were there other exclusion criteria other than not on mechanical ventilation (e.g. prior thoracic radiation, prior chemotherapy, others)?

Response:

The study included ALL patients (regardless of age) who had failed to benefit from conventional therapy and at risk for mechanical ventilation which in the LMIC carried a mortality of nearly 80% at the time. Pre-existing lung morbidity was an exclusion criterion, as were abnormal hematologic parameters below a defined threshold.

This has been revised in the manuscript. 

Refer to P. 7 lines 123-135.

2. It may be helpful to have subsections in italics (or separate paragraphs) for Cohort, Patient Assessment, Treatment (describing the LDRT), defining endpoints and then statistical/analytic methods.

Response:

We have included subsections in our revised manuscript

Refer to P. 6-10 lines 117-198.

3. P.14, line 114: Some more detail, even if brief, would be helpful. Planned in CT simulator, clinical setup on linear accelerator? Prescribed to isocenter, with or without heterogeneity corrections? Part of assessing feasibility includes the ability to deliver the treatment, so the authors should provide more information for the reader to assess the difficulty of accomplishing the protocol therapy. Lines 128-135 should be with line 114.

Response:

We have revised our manuscript and provided details regarding radiation planning and attempted to convey the technical difficulties associated with delivery RT in the LMIC under COVID 19 precautions.

Refer to P. 7-8 lines 144-166.

4. The authors should provide more detail on safety monitoring to meet the stated primary objective. How do you define/report feasibility? Hematologic toxicity is probably the primary immediate safety risk, given the large amount of bone marrow irradiated and likely including a lot of the spleen. Other trials have reported baseline and trends in hematologic indices, which would be helpful in this case. The authors focus more on vital signs, but presumably hematologic indices can also be reported since they were assessed (line 118).

Response:

The reviewer’s point is well taken to remind us of the potential for BM toxicity associated with RT. However, single LDRT in the literature has not been shown to cause BM suppression at the doses used. Figures 2 and 3 provide data supporting lack of clinically significant acute bone marrow suppression

Refer to P. 14 lines 234 & 237

5. Secondary objectives are reasonable, were these based upon other trials or developed prior to publication of other reports? It’s all a rapidly developing area, but it deserves some clarity here.

Response:

Secondary endpoints were based on previous studies as well clinically relevant parameters and that could be successfully accomplished within the constraints of the LMIC.

Results

1. P. 15, line 146: For a small trial, IQR may be less relevant than full range, especially given some concerns about younger patients treated. Consider reporting total range, not IQR.

Response:

The reviewer’s point is well taken and we have revised the manuscript to reflect the same. 

Refer to P.10 lines 201-202

2. P.16, line 156 and other tables: label each column by patient (PT) not LD for low dose treatment.

Response: 

Our revised manuscript has omitted the original Table 3 previously and provided the data in a block diagram (currently Figure 2 &3) which encompasses the relevant information in a more concise and easily readable format. All other tables have been formatted as suggested by the reviewers

3. P.16, line 156: Table 1 reports performance status at admission as 0. What scale is used, and if admitted presumably ill enough to not be a 0? Consider omitting.

Response:

We agree with the reviewers and have omitted the same from Table 1. 

4. P.16, line 156: Table 1 should report time from symptoms to LDRT. Consider moving performance status down in the table and including all temporal related metrics together.

Response:

We have revised Table 1 as per the suggestion by the reviewer and have added a column with time from symptom onset to LDRT within the manuscript. 

Refer to P. 11 Table 1

5. Baseline clinical parameters should include hematologic data.

Response:

The reviewers point is well taken and we have included two new figures (Figure 2&3) with information regarding the hematological data from the participants in the study.

Refer to P. 14 lines 234 & 237

6. P.17, line 162 Table 2: no description of ordinal score in text to permit interpretation.

Response:

Ordinal score has been defined in the Method section.

Refer to P.8&9 lines 171-176 

7. P. 18, lines 172-174 should be with the following paragraph, bringing all the radiation details together.

Response:

We have revised the flow of our revised manuscript as suggested by the reviewer

8. P.19, line 181: Table 3 – Need to include hematologic parameters as priority over inflammatory markers to demonstrate safety first. And consider reporting if toxicity recorded as yes/no or toxicity grade, rather than reporting numeric hematologic indices.

Response:

The reviewers’ point is well taken and we have omitted the table in place of box plots that provide the data in a more meaningful and understandable manner (Fig 2 and 3).

Refer to P. 14 lines 234 & 237

9. P.19, lines 181: Table 3 – what is the difference between “–“ and “n/a” – did some labs not get drawn? Patient 3 (LD3) has three empty cells. Needs clearer organization for the reader to interpret.

Response:

The reviewer’s point is well taken and we have omitted the table in place of box plots that provide the data in a more meaningful and understandable manner.

10. P.19, line 181: Table 3 – patient 10 is listed too early, biased by end outcome (death). Patients should be presented similarly through all Tables.

Response:

Table 3 has been omitted from the manuscript and replaced by box plots that reflect the inflammatory and hematological indices of the patients.

11. For the inflammatory markers of interest, consider graphing it in a figure, rather than tabular format.

Response:

We appreciate the reviewer’s comment and have omitted the table and now included Fig. 2 & 3 which provides the data in a box plot.

Refer to P. 14 lines 234 & 237

Discussion

1. P. 20, line 199: the Emory study was still essentially still a phase I, so didn’t evaluate effectiveness as much as safety.

Response:

We agree with the reviewer that our focus has to be feasibility and toxicity and that the clinical benefit a secondary, albeit important, observation. We have included recently published data from a phase II trial published from Emory in Dec 2020 that evaluated effectiveness.

Refer to P.15 line 259-264

2. The authors’ trial included a lot of patients with symptoms for a fairly long time and who required tocilizumab. It may be that the timing of intervention was too late in the pathophysiology of evolving COVID pneumonia. Despite the interest in LDRT, a randomized trial of intubated patients showed no clinical benefit. The authors need to put their trial cohort (younger, more advanced disease than Emory) on spectrum of severity compared to other trials. Part of the learning from these early trials will be defining an optimal cohort for phase II or III trials.

Response:

The reviewer makes an important point in that we need to put our trial in the context of other studies done in more developed countries. As previously mentioned, our patient population, while younger, was a sicker cohort than in the Emory (and other publications), and a majority had received Tocilizumab. We have revised our Discussion to encompass the reviewer’s point, including proposals for future clinical trials.

3. The authors should emphasize the safety of LDRT for acute effects, after clearly presenting them in Results. There also needs to be some acknowledgement of late effects as something beyond the scope of this trial that should be included in safety monitoring in future clinical trials.

Response:

We have now included hematologic toxicity data in Fig 2 & 3 and demonstrate no acute clinically significant hematologic toxicity. We agree that late toxicity are beyond the scope of our study but none of the 6 patients currently alive have presented with pulmonary or hematologic morbidity (data not included)

Refer to P. 18 Lines 314 & 315

4. Making the cost argument is valid. Given that treating patients once tocilizumab is needed for cytokine storm, should LDRT be considered earlier in the disease process for trials? It could result in cost savings if LDRT works. If not, why not?

Response:

The reviewer’s point is well taken and we have covered this suggestion across the Introduction and Discussion sections.

5. P.24, lines 270-272: This point on scarcity of linear accelerators needs to fit coherently from Introduction to Discussion. An opportunity to highlight need for investment to help cancer patients regardless of LDRT trial findings, made stronger if future studies of LDRT show a benefit.

Response:

We whole heartedly agree with the reviewer in that more RT units are needed in the LMIC, firstly to treat patients with cancer but in the new normal of Covid 19 could also LDRT. We have attempted to include this aspect in our Discussion.

Refer to P. 20 lines 340-344

Conclusion

1. Emphasize feasibility and acute safety.

Response:

We have revised the manuscript based on the suggestion by the reviewer. 

Refer to P. 20 lines 347-349

2. Recovery rate is can’t be attributed to LDRT success in line 277.

Response:

We agree with the reviewer and have omitted the same from the manuscript.

3. Lines 281-2: recommend being genetic as ‘drug therapy’ rather than specifying immune modulating agents. Other effective drug classes may also be too expensive.

Response:

The reviewer’s point is well taken and we have covered this suggestion across the manuscript

4. Lines 282-3: Final sentence not necessary

Response:

The reviewer’s point is well taken and we have omitted the sentence from the manuscript and also revised the Discussion.

Reviewer #2: 

The authors present the first study from African content on the use of LD-RT for COVID-19 ARDS. The study adds significant value to the current, limited understanding of the role of LD-RT for COVID-19 ARDS, and it's potential reproducibility of signal across other institutions across the world. Reproducibility of the signal across the world, in a different cohort of patients from Africa adds significant value, and is scientifically a notable accomplishment.

The authors also do a good job on only doing a small, feasibility study of 10 patients and asking the first level, simple safety and potential efficacy signal. Their patient selection of advanced, oxygen dependent patients is also reasonable. They chose patients that were not responding to other therapies and offered them the option of LD-RT and appears to find some clinical benefit. Their results of 6 patients improving is also in line with findings from others. There appears to be a correlation between inflammatory reductions post LD-RT and those patients that improve.

Minor edits:

1) Line 216, page 21. Did you mean to say 100 cGy instead of "gGy"?

2) Can the authors discuss pre LDRT rises in CRP, Ferritin, and other biomarkers and compare them with post LDRT changes? In the work by Hess et al, they saw significant rising trends preLD-RT, and then precipitous drops in some of these post LD-RT. Of these biomarkers, CRP appeared to be the most correlated. Did the authors see this in their own data sets (comparing pre/post LD-RT biomarker changes).. See Hess et al. Radiotherapy and Oncology. Vol 165. December 2021. Pages 20-31.

https://www.sciencedirect.com/science/article/pii/S0167814021087594?dgcid=rss_sd_all

3) The authors can also use the reference by Hess et al to add the comment that LD-RT could potentially have independently benefits patients, despite having gotten Remedesivir and Steroids.

4)The authors can add the following preclinical studies to also support their notion that LD-RT may potential help COVID-19 ARDS : (Meziani et al. Int. J. Radiation Oncology Biology and Physics. Volume 110. Issue 5. P 1283-1294. August 01, 2021). This is preclinical study looking at LD-RT in mouse using three different ARDS models and demonstrated that LD-RT helped convert inflammatory environment into an anti-inflammatory environment.

Second paper to considering adding: Jackson et al. Low Dose Radiotherapy for COVID-19 Lung Disease: Pre-clinical Efficacy in Bleomycin Model"... 2022. Jan 1; 112(1): 197-211. Int J. Radiation Oncology Biol & Physics. https://pubmed.ncbi.nlm.nih.gov/34478832/

Reviewer 2 combined Response:

We greatly appreciate the reviewer’s comments. Our revised manuscript includes the additional references provided and we are grateful for these additions, which serve to strengthen our submission. We have corrected the typo graphical error (Comment #1) and included a box plot for provides pre(Day 1)/post data on the relevant biomarkers (as suggested in comment #2). We are unable to include trends for inflammatory markers (Pre-LDRT) due to incomplete data from the patient medical records. We have also incorporate the input from Comment #3 and #4 including the references provided with relevant revision of the text. We are very grateful for the reviewers input and suggestion.

---

## [Decision Letter · Decision Letter 1]

26 May 2022

PONE-D-21-35607R1A Pilot Phase Ib/II Study of Whole-Lung Low Dose Radiation Therapy (LDRT) For the Treatment of Severe COVID-19 Pneumonia: First Experience from AfricaPLOS ONE

Dear Dr. Saleh,

Thank you for submitting your manuscript to PLOS ONE. After careful consideration, we feel that it has merit but does not fully meet PLOS ONE’s publication criteria as it currently stands. Therefore, we invite you to submit a revised version of the manuscript that addresses the points raised during the review process.

We look forward to receiving your revised manuscript.

Kind regards,

Randall J. Kimple

Academic Editor

PLOS ONE

Journal Requirements:

Additional Editor Comments (if provided):

I am quite sorry how long this has taken to review. Due to a policy change at PLOS One a statistical reviewer was added. If you can address the comments of reviewer 3 (which i hope will be straightforward) i am hopeful that we can relatively quickly proceed with this manuscript. Please also review the suggestions of reviewer 1.

Reviewers' comments:

Reviewer's Responses to Questions

**Comments to the Author**

1. If the authors have adequately addressed your comments raised in a previous round of review and you feel that this manuscript is now acceptable for publication, you may indicate that here to bypass the “Comments to the Author” section, enter your conflict of interest statement in the “Confidential to Editor” section, and submit your "Accept" recommendation.

Reviewer #1: All comments have been addressed

Reviewer #3: (No Response)

2. Is the manuscript technically sound, and do the data support the conclusions?

Reviewer #1: Yes

Reviewer #3: Yes

3. Has the statistical analysis been performed appropriately and rigorously? 

Reviewer #1: Yes

Reviewer #3: Yes

4. Have the authors made all data underlying the findings in their manuscript fully available?

Reviewer #1: Yes

Reviewer #3: Yes

5. Is the manuscript presented in an intelligible fashion and written in standard English?

Reviewer #1: Yes

Reviewer #3: Yes

6. Review Comments to the Author

Reviewer #1: Thank you to the authors for resubmitting this manuscript. It has improved substantially and reads well. only two stylistic points to consider:

1. Lines 88-89 - consider citing the other studies here but leave that to author preference

2. Line 306: 'discharge' would be better as past tense 'discharged'.

Thank you again for the opportunity to review this interesting work.

Reviewer #3: Nicely written paper, I have some minor comments:

Why is no trial registration number reported?

Replace the figures in figure 3 with spaghetti-plots (i.e. individual lines per patient), "dead" or "Discharged" can be different line types but each patient can have a different colour. By doing this with figure 3 it will complement figure 2 (which shows the distribution at each time point) with the individual progress per patient over time.

7. PLOS authors have the option to publish the peer review history of their article (what does this mean?). If published, this will include your full peer review and any attached files.

Reviewer #1: **Yes: **Matthew S Katz, MD

Reviewer #3: No

---

## [Author Response · Author response to Decision Letter 1]

4 Jun 2022

A Pilot Phase Ib/II Study of Whole-Lung Low Dose Radiation Therapy (LDRT) For the Treatment of Severe COVID-19 Pneumonia: First Experience from Africa

Dear Dr Kimple,

We very much appreciate the review of our revised manuscript and the very helpful comments provided by the reviewer. We have responded to each of the comments and revised our manuscript accordingly. The revised manuscript includes the revisions as well as additional references. We thank the reviewers for their comments for which has served to strengthen our submission. We look forward to a positive decision by the editorial team.

Herewith please find our responses to each of the reviewer comments/suggestions:

Reviewer #1: 

Comment:

Thank you to the authors for resubmitting this manuscript. It has improved substantially and reads well. only two stylistic points to consider:

1. Lines 88-89 - consider citing the other studies here but leave that to author preference

2. Line 306: 'discharge' would be better as past tense 'discharged'.

Response:

The reviewer’s insight and input is much appreciated and we have revised our manuscript and incorporated these suggestions.

Refer to P. 5 Line 88-89 and P. 18 Line 313(previously line 306)

Reviewer #3: 

Comment:

Nicely written paper, I have some minor comments:

1. Why is no trial registration number reported?

Response:

The reviewer’s comments are well received and we have now included the trial registration number in the manuscript. The trial registration number had been previously submitted via the submission portal during the original submission.

Refer to P. 6 Line 109-110

2. Replace the figures in figure 3 with spaghetti-plots (i.e. individual lines per patient), "dead" or "Discharged" can be different line types but each patient can have a different colour. By doing this with figure 3 it will complement figure 2 (which shows the distribution at each time point) with the individual progress per patient over time.

Response:

We very much appreciate the insightful input from Reviewer # 3 and the suggestion to use a spaghetti plot as opposed to the box plot for Fig. 3. We have followed this advice. However, the small sample size and missing data points for some of the patients makes the spaghetti plot less definitive in demonstrating the upward trend in inflammatory markers in those patients who died. This trend seems to be more apparent when data is summarized into a boxplot for each inflammatory marker. For this reason, we have opted to include the original boxplots as supplementary material. 

Refer to Figure 3 Page 14 Line 238-239, 250-255 and Page 18 Line 328-331

---

## [Editor Report · Decision Letter 2]

14 Jun 2022

A Pilot Phase Ib/II Study of Whole-Lung Low Dose Radiation Therapy (LDRT) For the Treatment of Severe COVID-19 Pneumonia: First Experience from Africa

PONE-D-21-35607R2

Dear Dr. Saleh,

We’re pleased to inform you that your manuscript has been judged scientifically suitable for publication and will be formally accepted for publication once it meets all outstanding technical requirements.

Kind regards,

Randall J. Kimple

Academic Editor

PLOS ONE

Additional Editor Comments (optional):

thank you for your patience with the review process for this manuscript.
---

## [Editor Report · Acceptance letter]

23 Jun 2022

PONE-D-21-35607R2 

A Pilot Phase Ib/II Study of Whole-Lung Low Dose Radiation Therapy (LDRT) For the Treatment of Severe COVID-19 Pneumonia: First Experience from Africa 

Dear Dr. Saleh:

I'm pleased to inform you that your manuscript has been deemed suitable for publication in PLOS ONE. Congratulations! Your manuscript is now with our production department. 

Kind regards, 

on behalf of

Dr. Randall J. Kimple 

Academic Editor

PLOS ONE